# Analog Computing for AI Sometimes Needs Correction by Digital Computing: Why and When

**Changdae Kim, Daegun Yoon, Taehoon Kim, Yeonjeong Jeong,**
**Kangho Kim, Kwangwon Koh, Eunji Pak**
Electronics and Telecommunications Research Institute (ETRI)
cdkim, kljp, taehoon.kim, yjjeong, khk, kwangwon.koh, pakeunji@etri.re.kr

## Abstract

Analog computing is a compute-in-memory technology that allows multiple dot-product operations to be performed in parallel, and can extremely accelerate matrix-vector multiplications in AI. However, computation in analog computing devices is imperfect due to its non-idealities. This can severely degrade its accuracy, and prevents analog computing from being widely used.

In this paper, we propose a correction-based approach to mitigate these non-idealities of analog computing. The proposed method exploits the confidence score calculated from analog computing, and leverages digital computing to correct the result if the confidence is low. We propose two algorithms that efficiently improve accuracy without requiring offline profiling or training. First, the digital rate based correction algorithm optimizes the accuracy within the target digital computing usage rate. Second, the confidence threshold based correction algorithm balances the digital computing usage rate and the accuracy by finding the appropriate threshold from online profiling. We use several image recognition models to show that the proposed algorithms improve the accuracy of AI in analog computing by efficiently utilizing digital computing.

## 1 Introduction

Analog computing is a compute-in-memory technology based on the law of physics. An analog computing device has crossbars of memory cells like other memory devices. If voltages are applied to one side, the results of dot-products between these voltages and the conductance stored in the cells are derived using Ohm's and Kirchhoff's laws. This phenomenon allows Matrix-Vector Multiplication (MVM), a key operation in AI models, to be performed in $O(1)$ time complexity. In addition, since the matrix elements reside in the crossbars as the conductance during computation, the memory bandwidth requirements for MVM operations are significantly reduced. These advantages make analog computing a promising solution to extremely improve the performance and energy efficiency of AI, compared to digital computing devices, like GPUs or NPUs.

However, analog computing introduces computational noise due to its non-idealities, resulting in imperfect results and limiting its adoption. There are many sources of computational noises. First, noise can be introduced when converting input value to voltage level. Second, the matrix elements need to be programmed as the conductance level of the memory cells, but it is difficult to fine-tune the conductance level. Moreover, the conductance can leak over time. Third, the dot-product result is derived as the current level, which can be affected by electrical interference or wire capacity limitations. Fourth, converting the current level to digital output is prone to errors, and its resolution may be limited.

To mitigate these non-idealities, several approaches have been proposed. Some works exploit redundancy when storing or processing data to improve computational accuracy [1, 6]. Several archi-

Second Workshop on Machine Learning with New Compute Paradigms at NeurIPS 2024(MLNCP 2024).

tectural studies have been conducted [24, 4]. For example, a large shared analog-to-digital converter (ADC) enhances bit resolution without overhead [24]. Some works optimize AI models for analog devices, such as analog-aware training to mitigate the conductance leakage effect [21, 26] and tailored stochastic gradient descent (SGD) algorithm [27]. In addition, hybrid analog-digital architectures have been proposed [9, 13, 15, 17, 18]. In these architectures, digital computing handles the accuracy-sensitive parts while analog computing handles the rest to increase energy efficiency.

In this paper, we propose a correction-based approach to overcome the non-ideality of analog computing. In our approach, we first use analog computing devices to perform AI models and check the confidence of the result. Then, if the result is not confident enough, we correct it with digital computing. To determine confidence, we use the difference between the maximum and the second maximum softmax value [23]. We propose two algorithms for the correction. The first algorithm improves the accuracy of the AI model while maintaining the target digital computing device usage rate. The second algorithm determines the appropriate confidence threshold from the profiling phase. Then, in the running phase, the threshold is used to determine whether the result should be corrected. The proposed algorithms do not require any offline profiling or training.

We show the feasibility of the proposed mechanisms with an analog computing device emulator [22]. With four image classification AI models, we achieve near digital accuracy, with only 22%-62% support of digital computing. The proposed mechanisms are independent of the hardware architecture and can be used with any AI model that ends with a softmax model, including LLMs. In future work, we will apply the proposed mechanisms to other AI models including LLMs, and analyze the latency and power consumption with the detailed hardware simulation.

## 2 Background and Related Work

### 2.1 Analog Computing and Its Non-Idealities

The operating principle of analog computing is shown in Figure 1. Here's how it works: if we program the conductance of each memory cell as $G_{i,j}$, and then apply a voltage $V_i$ to each row, then according to Ohm's law and Kirchhoff's law, the current flowing through the $j$-th column is equivalent to $\sum_i G_{ji} \cdot V_i$. Based on this principle, matrix-vector multiplications can be implemented in a fully parallel fashion, in a single time step.

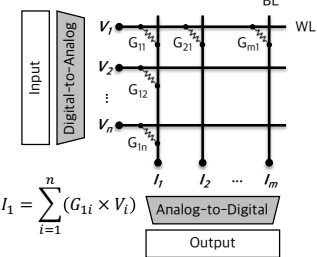

As mentioned above, analog computing offers great efficiency through highly parallelized, in-situ computations. It has fast computation speed and high energy efficiency. In addition, it alleviates the memory bandwidth bottleneck by performing computations where the data resides.

Figure 1: Analog computing

However, analog computing is inherently vulnerable thus it is difficult to achieve the same level of computational accuracy as digital computing. Analog devices in nature have programming noise, read noise, temporal resistance drift, limited bit-level precision, analog-digital conversion loss, etc [3, 4, 11, 19, 24, 26]. These non-ideal factors might degrade computational accuracy. Therefore, for practical use, it is important to maintain a certain level of accuracy in AI computation under these non-idealities of analog computation.

### 2.2 Related Works to Overcome Non-Idealities

To alleviate the accuracy degradation of AI computation caused by non-ideality, many previous works have been proposed. [4, 24, 26] proposed novel architectural designs that can minimize the impact of conversion loss bit-level precision limitation and analog-digital conversion loss. [1, 6] presented a bit-level divide-and-merge mechanism to recover from the bit-level precision limit. [16] proposed a time-domain conversion technique and buffer for storing intermediate analog values to minimize losses in analog-digital conversion. [21, 26] proposed a new AI training algorithm that is robust to temporal conductance drift. Some work [12, 19] aimed to minimize the stuck-at faults using redundant arrays and re-mapping weights. Tiki-Taka [27] addressed the inefficiency and high error rates of stochastic gradient descent (SGD) on analog devices using an auxiliary array.

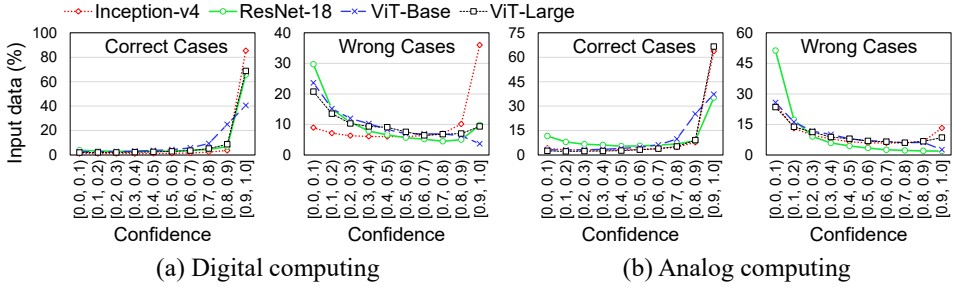

Figure 2: Confidence score and actual correctness

On the other hand, several works introduced hybrid designs that combines analog computing with highly accurate digital computing to compensate for non-ideality. Rashed et al. [9] proposed an architecture that compute the most significant bits (MSBs) on digital logic and the least significant bits (LSBs) on analog crossbars. Nandakumar et al. [18] presented a design where high-precision digital computing accumulates the partial sum of the results calculated in an analog computing device and updates the precise weights in the analog computing device. ReHy [13] delegated feed-forward propagation to analog computing, while backward propagation, which requires precise computation, is handled by digital computing for CNN training. 3D-ReG [15] is a heterogeneous architecture that vertically integrates ReRAM-based analog device and GPU to improve DNN training efficiency. HARDSEA [17] aimed to implement self-attention in transformer models using a hybrid design that offloads dense relevance predictions to analog computing devices and handles sparse, precise computations with digital computing.

## 3 Design

In this paper, we propose analog-digital hybrid AI computation that, only when the AI computation in analog computing devices seems to be wrong, digital device steps and corrects the answer. This approach takes advantage of both the high efficiency of analog computing and the high accuracy of digital computing.

The problems we propose to solve in our approach are: (1) How do we know that an analog AI computation is wrong? In the real world, the correct answer is unknown. (2) How does digital computing decide whether to intervene? We need a strategy that balances analog computing's efficiency and digital computing's correctness. In this section, we discuss the answers to both questions.

### 3.1 Confidence and Correctness in AI Models

To predict the correctness of analog computation, we use the confidence score of the AI model. There are several ways to define a confidence score [23], and we choose the difference between the top two softmax values from the last layer. The softmax values from the last layer represent the likelihood of each item or class being selected as the output, so the difference between the top two softmax values represents the confidence of the best answer compared to the second best answer.

The principle behind our approach is that when the difference between the top two softmax values is sufficiently large, there is little chance that their order will be reversed due to analog calculation errors. Figure 2(a) presents the relationship between confidence score and actual correctness of four AI models run on a digital device. It has two plots: the left with data from accurate predictions, and the right from incorrect ones. The x-axis is the confidence score range, and the y-axis indicates the number of cases that their data have the confidence score belonging to the range in the x-axis. These plots demonstrate a clear trend: accurate predictions are likely to have higher confidence score, and wrong predictions usually have lower confidence score. Figure 2(b) shows results when using analog computing, emulated through IBM AIHWKIT [22], with further details provided in Section 4. In analog computing, the trend between confidence and accuracy remains consistent.

To address computational errors in analog computing devices, our solution involves initially performing the computations on the analog device. If confidence is sufficiently high, we accept the result; otherwise, the result is corrected by the digital accelerators. Obtaining confidence score is straightforward: the AI model first identifies the maximum softmax value, and during this process,

the second highest softmax value is naturally obtained with little extra effort. To offload the computation, we can take advantage of the hybrid approach by transferring only the input data, which is likely to reside in the host memory, to the digital accelerators.

We propose two mechanisms for determining a confidence threshold for triggering digital device intervention. The threshold is determined based on either (1) the target utilization rate of digital computation, or (2) continuous online profiling. Both mechanisms are based on online profiling, not requiring offline profiling. This enables immediate adaptation without pre-processing AI model on both analog and digital computing devices to collect data in advance.

## 3.2 Digital-Rate based Correction Mechanism

In this method, the system administrator determines the target utilization rate of digital computation, $\alpha\%$, taking into account the available resources and power conditions. The AI model initially processes incoming data on an analog device first, generating a confidence score and a result. In case the confidence score ranks within the lowest $\alpha\%$, the computation is transferred to a digital accelerator, which reprocesses the input and updates the result accordingly. This approach primarily uses analog computing, engaging digital computing only for the least confident $\alpha\%$ cases.

---

**Algorithm 1** Digital-Rate based Correction Algorithm

---

1: Compute the AI model with analog computing device
2: $idx \leftarrow$ **round_down**($confidence\_score/Histogram.interval$)
3: $Histogram.bin[idx] \leftarrow Histogram.bin[idx] + 1$
4: $Histogram.total \leftarrow Histogram.total + 1$
5: $rank \leftarrow \left( \sum_{i=0}^{idx-1} Histogram.bin[i] \right) +$ **randint**($0, Histogram.bin[idx]$)
6: **if** $rank/Histogram.total < digital\_rate$ **then**
7:     Correct the result with digital computing device
8: **end if**

---

Algorithm 1 describes the overall procedure. An online histogram tracks the distribution of confidence, from which the rank of a specific confidence is determined. Assuming the confidence falls into bin index $idx$, then its rank is $\sum_{i=0}^{idx-1}$ **count_of_bin**$(idx) +$ **randint**$(0, $**count_of_bin**$(k))$

From the calculated rank, we can get the percentile of the confidence. When the percentile is lower than $\alpha$, we re-run the AI model in a digital accelerator with same input and update the result.

## 3.3 Confidence Threshold based Correction Mechanism

This mechanism is designed to determine the optimal confidence threshold through online profiling. It is intended to meet scenarios in that administrators do not need to set a digital computing usage rate or prefer the system to manage settings autonomously.

---

**Algorithm 2** Confidence Threshold base Correction Algorithm

---

1: $data\_id \leftarrow data\_id + 1$
2: **if** $data\_id\%phase\_len < prof\_rate \times phase\_len$ **then**            ▷ Profiling phase
3:     Compute the AI model with analog and digital computing devices in parallel
4:     **if** $Result_{analog} \neq Result_{digital}$ **then**
5:         $profile$.**append**(confidence score from analog computing)
6:     **end if**
7: **else**
8:     **if** $data\_id\%phase\_len == prof\_rate \times phase\_len$ **then**     ▷ Start of execution phase
9:         $threshold' = \delta\%th$ percentile of profiled confidence scores
10:         $threshold \leftarrow threshold \times \beta + threshold' \times (1 - \beta)$
11:     **end if**
12:     Compute the AI model with analog computing device
13:     **if** $confidence\_score \leq threshold$ **then**
14:         Correct the result with digital computing device
15:     **end if**
16: **end if**

---

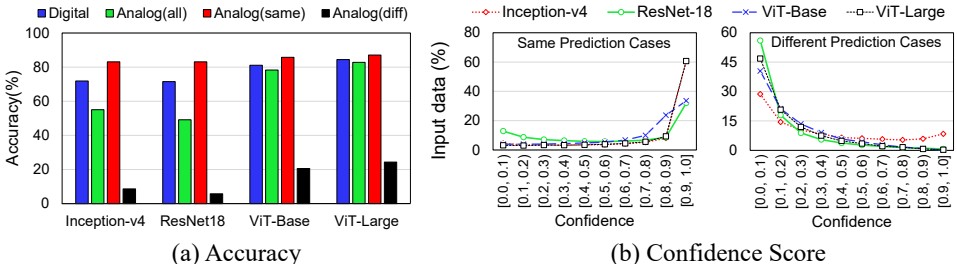

|  |  |
| :---: | :---: |
| (a) Accuracy | (b) Confidence Score |

Figure 3: Accuracy and confidence score when the prediction results from digital and analog computings are same or different

Algorithm 2 describes the entire procedure, which consists of two phases: profiling and execution. In the profiling phase, input data is simultaneously processed by both analog and digital computing devices to collect the distribution of confidence. At the end of the profiling phase, a threshold is established. In the execution phase, the AI model is executed on analog computing devices and the results are adjusted by digital computing if the confidence is less than or equal to the threshold value.

The key here is to determine the appropriate threshold. If the threshold is set too low, digital computing may be unable to calibrate the results, potentially degrading the accuracy of AI results. Conversely, if the threshold is set too high, excessive digital computing will be employed, which could undermine the efficiency of analog computing.

To determine the threshold, we specifically profile cases where the analog and digital computing results differ. As digital computation guarantees the correctness, its accuracy is the upper limit for a given model. Thus, the discrepancy between the two results indicates that the analog result is likely to be wrong. Figure 3 (a) shows the accuracy of digital computing, analog computing, and analog computing when the prediction results are the same or different with digital computing. This indicates that the different prediction cases are the main cause of low accuracy of analog computing. Note that the same prediction cases have larger confidence scores. They are easier to pick the right answer. In addition, Figure 3 (b) shows that confidence score distribution is highly skewed for the same and different prediction cases. Therefore, we profile different prediction cases to determine the range of the *'confidence score that needs to be corrected'*. We set the threshold with two different options, the 70-th and 90-th percentile of these profiled confidence scores. In the algorithm, $\delta$ represents the percentile value. Profiling is conducted periodically, and the threshold is updated using a weighted moving average, with a weight of 0.5. $\beta$ in the algorithm denotes this weight.

It is possible that no cases are found in which the results of analog and digital computing differ. In such a case, we set the confidence threshold for profiling to 0. This is a reasonable assumption because it is highly probable that the results of analog and digital computing will be identical.

## 4 Initial Results

### 4.1 Experimental Methodology

In this paper, we focus on evaluating the accuracy of AI computation and the usage of digital computing. Accuracy is the key target metric, and digital computing usage can indicate the achievable energy efficiency and performance improvements of the proposed mechanisms. We leave a detailed analysis of latency and power consumption as future work.

To emulate computational noises on analog computing devices, we utilize the IBM AIHWKIT [22]. We mostly use default configurations: PCM-like memory cells, 512×512 crossbar arrays, 8-bit DAC, and 8-bit ADC. Note that our mechanism is orthogonal to the characteristics of analog computing devices, and evaluation with various device configurations is our on-going work.

To evaluate the accuracy of analog computing devices, we choose the image classification tasks as it is easy to verify the correctness with labeled dataset. We use four image classification models, ResNet-18 [10], Inception-v4 [25], ViT-Base, and ViT-Large [8]. For the dataset, we use ImageNet data [7] since it has a large number of classes. In addition, it has 50,000 images with labels for validation. We use the validation set as input data for evaluation.

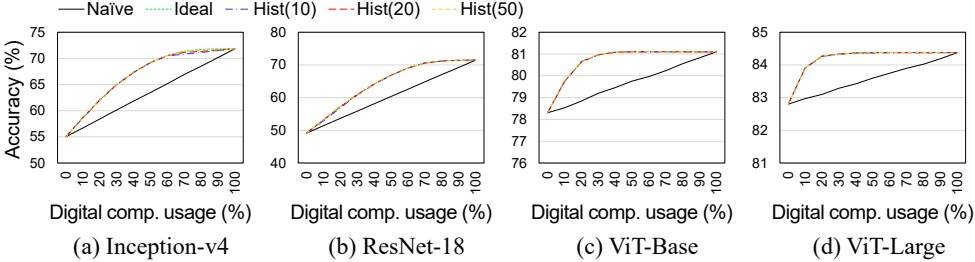

(a) Inception-v4     (b) ResNet-18     (c) ViT-Base     (d) ViT-Large

Figure 4: Accuracy and digital computing usage with digital-rate based correction

## 4.2 Results with Digital-Rate based Correction

Figure 4 shows the results with digital-rate based correction algorithm. The x-axis represents the target digital computing usage rate, where 0% means all computation is performed on analog computing devices, and 100% indicates only digital computing is used. The y-axis shows the accuracy of the image classification task.

For comparison, we define *Naïve* algorithm, which randomly selects input data to correct its result with digital computing. *Ideal* algorithm uses knowledge of all future input data. Finally, *Hist(k)* algorithms refer to the instance of Algorithm 1, with $k$ indicating the number of histogram bins.

The four graphs with different models show similar trends, despite varying accuracy on analog and digital computing devices. *Naïve* shows a linear improvement in accuracy as the digital computing rate increases, while *Ideal* shows a super-linear improvement. This confirms that the confidence score based correction mechanism effectively identifies potentially erroneous inputs in analog computing. Furthermore, there is almost no gap between *Ideal* and *Hist(k)*, indicating that the histogram based online profiling accurately predicts the confidence rank of the result. We also verified that actual digital computing usage is close to the target rate, but the graph is omitted due to space limits.

## 4.3 Results with Confidence Threshold based Correction

Figure 5 presents the evaluation results of confidence threshold based correction algorithm. The x-axis is the different comparisons. For reference, we present the result of *Analog-only* and *Digital-only* computation. The *Ideal* is the case where the threshold is determined with the complete knowledge of all results for the entire dataset. *Profile($\rho$%)* are the instances of Algorithm 2 where $\rho$ is the profiling rate. Each epoch consists of 1,000 inputs; for a 10% profiling rate, 100 images are used for profiling, followed by 900 in the execution phase. In the Figure, the lines represent the accuracy (y-axis on the left side) and the bars show the digital computing usage rate (y-axis on the right side). The green line and bars represent results when we determine the threshold as the 70th percentile of profiled confidence scores. The blue line and blue bars use the 90th percentile.

Four graphs show similar trends. The proposed algorithm achieves accuracy close to that of *digital-only* computation while increasing the usage of analog computing to 38-78%, thereby enhancing computing efficiency. The profiling-based threshold setting algorithm performs comparably to the *ideal* case, suggesting that online profiling is sufficient, even with just a 5% profiling rate. Increasing the profiling rate may have better results, however, it is less efficient since both analog and digital computations are required during profiling. Comparing the green and blue graphs, a trade-off between accuracy and efficiency is observed when setting the threshold at the 70th and 90th percentile, with the 90th percentile providing more achievable results.

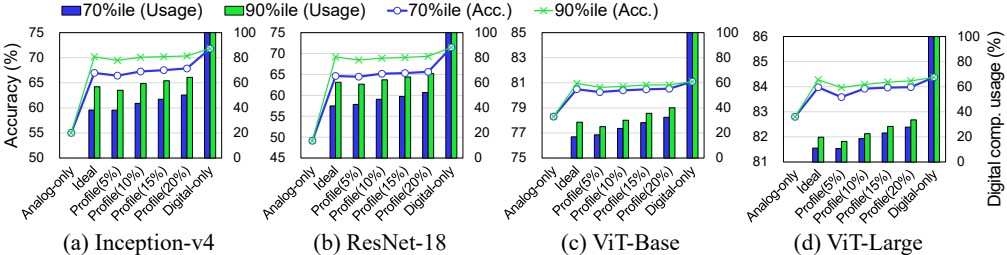

(a) Inception-v4     (b) ResNet-18     (c) ViT-Base     (d) ViT-Large

Figure 5: Accuracy and digital computing usage with confident threshold based correction

# 5 Discussion and Conclusion

In this paper, we propose a correction-based approach to overcome analog computing's non-ideality. We find that the confidence from the result of the AI model can be a good indicator of its correctness, and improve the overall accuracy while minimizing digital computing usage. Our early experimental results show that the proposed algorithms based on confidence score efficiently optimize the digital computing usage to achieve accuracy close to digital computing.

One important advantage of the proposed mechanism is that it is both hardware and model independent. Although our initial focus has been on analog computing devices with crossbar arrays, this mechanism can be applied to any hardware experiencing inherent computational noises, e.g., photonic computing [5]. Furthermore, it can be applied to any AI model that makes a selection in the final layer. This includes most recognition models, which select one class at the last layer, and most generative models, which choose the next token or the color of the next pixel at the last layer.

Our future work is as follows. First, detailed hardware simulation is required to prove the efficiency of the proposed algorithms. Especially, the latency and energy efficiency are affected by hardware configurations of analog computing. We will evaluate our mechanisms with various analog computing devices using the detailed analog hardware simulator, e.g., DNN+NeuroSim [14, 20].

Second, we can apply the proposed methods to other AI models, including Large Language Models (LLMs). Since the confidence based mechanisms have been used for LLMs [2, 23], we believe that our approach is easily adopted for LLMs. If the capacity of the analog computing device is limited, analog computing can be applied to only the initial layers or the distillation models.

Third, we will analyze the combined effect of other methods designed to overcome the non-idealities of analog computing. For instance, hardware-aware training [21, 26] can enhance the computation accuracy of analog computing and reduce the digital computing requirements when combined with the proposed mechanism.

## Acknowledgments and Disclosure of Funding

This work was supported by Institute of Information & communications Technology Planning & Evaluation (IITP) grant funded by the Korea government(MSIT) (No.RS-2023-00216370, Research on High-efficiency Analog AI Computing for Large-scale DNN Model).

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
