# OpenReview forum: "Analog Computing for AI Sometimes Needs Correction by Digital Computing: Why and When"
_NeurIPS.cc/2024/Workshop/MLNCP — MLNCP Poster_

### Official Review · Reviewer_wPN5 · 2024-10-01
**The authors introduce two algorithms aimed at approving the accuracy of analog hardware by offloading a fraction of the computation to digital hardware**

**Rating:** 7
**Confidence:** 3

**Review:**

This paper presents two methods to remedy the reduced accuracy of analog hardware caused by its non-idealities. Both methods are based on estimating the confidence of an analog neural networks output, and then correcting low-confidence outputs by performing an additional run with a digital-only neural network. The authors test these methods by using a employing a software library that simulates analog electroinic crossbar arrays. Results indicate that the accuracy could be improved by up to 20 percentage points (depending on the model and method used). Overall, I think this is a nicely written paper and I am in favor of accepting it; I just have some minor comments and questions:

- The authors use analog computing pretty much synonymously with analog electronic crossbar arrays. For completeness, I think it should be mentioned that analog computing also encompasses optical or photonic computing (and potentially even mechanical or other forms of computing, though these are rather niche). Since their method is hardware and model agnostic, it should also be applicable to other kinds of hardware.
- It would be interesting to learn what happens if one combines the correction-based approach presented here with other methods aimed at overcoming analog hardware non-idealities such as hardware aware training. Would the combined approach increase the accuracy even further or would the improvements be neglegible?
- As the authors themselves point out, applying the proposed methods to other AI models would be very illuminating. Even the 4 models shown in this paper exhibit significant variation in how much digital computation is needed to get an accuracy close to that of digital hardware.
- In Algorithms 1 and 2, the variable names could be more clearly explained in the text. For example, the variable $\beta$ is introduced in line 10 in Algorithm 2 without explanation of what it means. From context, one can infer that it is the weight of the moving average, but this is never explicitly said in the text. In fact, in the main text the weight of the moving average is said to be equal to 0.5 always. Adding the definition of these variables would make the algorithms a lot more readable.
- There is a typo on line 168: It should be either _no case is found_ or _no cases are found_.

---

### Official Review · Reviewer_afrQ · 2024-10-03
**Review for Analog Computing for AI Sometimes Needs Correction by Digital Computing: Why and When**

**Rating:** 7
**Confidence:** 3

**Review:**

The paper presents an algorithm to automatically coordinate analog and digital compute to balance the power efficiency and speed benefits of analog computing with the noise resiliency and accuracy of digital computing. Specifically, the authors propose to execute a digital model when needed based on the confidence of the output provided by the analog model. The confidence of the analog model is estimated by the distribution of softmax outputs, and thus, this methodology could be generally applied to many types of networks including LLMs.

Implicitly this approach relies on there being a connection between instances where the analog network deviates from the digital network and instances where the outputs of the analog network have low confidence. While the intuitions described in section 3.1 make sense, it is difficult to conceptualize the distribution of error at the output of the analog network. It would be great if additional quantitative analysis could be provided to highlight the degree to which the chosen confidence metric predicts deviations between the output of the analog model and outputs of the digital model. Specifically, while figure 2 shows how confidence reflects when digital and analog outputs are incorrect, it does not seem to show how confidence reflects when digital and analog outputs are different, and that is the challenge that is resolved by the proposed method.

Other than this, the paper is well-written and effective in discussing a couple intuitive strategies for choosing when to execute the digital model. Additionally, the authors experimentally validate their algorithm and demonstrate digital-like performance with reduced computation. It would also be good to include more discussion on how future work could potentially provide more precise estimates of when the analog and digital models deviate in order to further limit the amount of digital network executions.

---

### Decision · Program_Chairs · 2024-10-10

Accept (Poster)